# The Impact of Male Social Status on Vaginal Secretory Responses in Mice

**DOI:** 10.3390/biology14081041

**Published:** 2025-08-13

**Authors:** Natalia Murataeva, Sam Mattox, Alex Straiker

**Affiliations:** 1Department of Psychological and Brain Sciences, Indiana University, Bloomington, IN 47405, USA; nmuratae@iu.edu (N.M.); sammatto@iu.edu (S.M.); 2Gill Institute for Neuroscience, Indiana University, Bloomington, IN 47405, USA

**Keywords:** vaginal, pheromone, chemical messenger, behavioral ecology, social status, attraction

## Abstract

A recently described murine model of vaginal secretory responses to male pheromones offers a window into how mice determine the likelihood of coitus and the chemical messengers that mediate some of these responses. Here we show that female mice experience a vaginal response to the scent of dominant but not subordinate males. In a second finding, we find that in contrast to farnesenes, a second candidate volatile preputial gland-derived chemical messenger 1-hexadecanol does not elicit a response. These findings narrow both the range of conditions and the chemical messengers that may elicit a coital preparatory response in laboratory mice.

## 1. Introduction

Mice rely heavily on chemical messengers to communicate key information relating to their identity, social status, emotional state, and health (reviewed in [1,2,3]). Female mice can distinguish the social status of males by scent and spend more time investigating the scent of dominant relative to subordinate males [4]. Urine of male mice carries factors that communicate their social status, particularly α and β farnesenes, volatile terpenic messengers that are produced in the male preputial gland [5]. Brain et al. [6] showed that preputial glands of subordinate males appear to be less well developed. Harvey et al. similarly showed that preputial glands were smaller in subordinate males and additionally showed profound differences in levels of α/β farnesenes, with levels much higher in dominant males relative to subordinate males [7]. If subordinate males produce less of the farnesenes, this may translate to different physiological responses in females. We have recently described a murine model of vaginal secretion that allows the measurement of minute changes in vaginal secretion in laboratory mice [8]. Exposure to the odor of male mice was sufficient to induce a robust vaginal secretory response. Such a response would serve to prepare the reproductive apparatus at the prospect of coitus (e.g., [9]). This model is well-suited to test whether or not reduced chemical messenger production in subordinate males translates into a reduced vaginal secretory response. This has not previously been tested and therefore represents a gap in knowledge. Our hypothesis is that subordinate males will elicit a reduced vaginal response relative to dominant males.

A related question has to do with the chemical messengers themselves. Is there a one-to-one relationship between a given messenger and the message or can different messengers communicate the same message? In the case of attractants, several messengers with different origins and chemical properties have been proposed, based largely on investigation times in the face of scent choice. This has provided important insights into rodent behavior, but there is an open question as to whether or not a compound determined to be an attractant in this way will also elicit a vaginal secretory response. In this model we have shown that the vaginal preparatory response can be induced by the preputial gland-derived volatile chemical messengers α/β farnesenes [8,10]. But other messengers have been identified as female attractants, including 1-hexadecanol [11,12,13] and the non-volatile major urinary protein darcin [14], as well as 3,4-dehydro-exo-brevicomin and 2-sec-butyl-4,5-dihydrothiazole (SBT) [15]. Of these, 1-hexadecanol is, like the farnesenes, proposed to be both volatile and derived from the male preputial glands. We have demonstrated the effectiveness of farnesenes in eliciting a vaginal secretory response but 1-hexadecanol, a second volatile preputial gland-derived proposed pheromone, has not been tested, representing a second gap in knowledge. Our hypothesis is that if 1-hexadecanol is an attractant pheromone it will similarly evoke a vaginal response.

In the present study, the central aim is to make use of a previously unavailable assay to test physiological vaginal secretory responses to the prospect of coitus. We tested whether or not the social status of males impacts the vaginal secretory responses of females, and additionally tested for a secretory response to the putative female attractant 1-hexadecanol.

## 2. Materials and Methods

### 2.1. Study Animals

Mice were group-housed in standard ventilated caging (cage floor dimensions: 33.5 × 18 cm, depth 14 cm), with corn cob-based bedding (Bed-o’combs laboratory animal bedding, The Andersons, Maumee, OH, USA). Mice were group-housed 3–4 mice per cage and were fed Inotiv Teklad 2918 irradiated rodent diet ad libitum. Because mice are nocturnal, we tested their responses during their active phase by maintaining them on a reverse light cycle. If moved from a different light cycle, mice were given 2 weeks to acclimatize. CD1 strain mice were used, partly because we have found that, in contrast to the C57BL/6 strain, these mice appear to have an intact circadian response [8]. The mice were bred in a colony in the same facility and kindly provided by Dr. Ken Mackie (Indiana University, Bloomington, IN, USA). Only female mice were tested, though urine from males was used as an odorant. Animals were returned to the colony after the experiments. All animal care and experimental procedures used in this study were approved by the Institutional Animal Care and Use Committee of Indiana University (protocol# 24-030, approval date 3 September 2024) and conformed to the Guidelines of the National Institutes of Health on the Care and Use of Laboratory Animals. Experiments complied with ARRIVE guidelines.

### 2.2. Sex and Number of Mice Used for Each Experiment

Female mice (age 2–6 months) were tested for vaginal responses. This age range was used because females are readily able to mate and bear young during this age range and because the vaginal secretory response is a robust physiological response that is essential for procreation. Males provided urine samples. Dominant and subordinate males were identified based on relative weight within a given cage (reviewed in [16]). In each cage of 3–4 males (age 3–6 months), the heaviest and lightest male was taken as dominant and subordinate, respectively, if there was at least a 10 g difference in their weights.

Total number of mice used for these experiments: 98.

Responses to dominant males: 10 females.

Responses to subordinate males: 9 females (repeated with proestrus/estrus only: 8 females).

Scent investigation dominant vs. subordinate: 21 females.

Responses to 1-hexadecanol: 21 females (repeated with proestrus/estrus only: 13 females).

Responses to 1-hexadecanol (physical contact): 9 females.

Scent investigation time 1-hexadecanol: 7 females.

Sample sizes were determined in our previous study introducing this method of measuring vaginal secretory function [8]. A specific randomization paradigm was not employed to allocate mice for experiments.

### 2.3. Measurement of Vaginal Moisture

We have recently described the use of colorimetric threads for measurement of vaginal moisture as well as the method of manufacture of these threads [8]. An image of the cannulated thread and a sample stained thread for measurement can be seen in Figure 1 of Andreis et al., 2022 [17]. This method is brief (10 s of measurement), minimally invasive, well-tolerated by the mice, and does not require anesthesia. The 10 s duration results in the measurement of clear differences under various experimental conditions, as outlined in [8]. Briefly, to allow for consistent placement in the vaginal cavity, a phenol red-coated thread is inserted into a glass capillary (A-M Systems, 0.86 mm inner diameter, Cat#:593200, fire-polished to prevent injury), leaving 3 mm of thread outside the opening of the capillary. The capillary is then placed into the vaginal cavity of an unanesthetized mouse for 10 s [8]. The distance that liquid from the vaginal cavity travels along the thread is quantified and taken as a measure of vaginal moisture, as described previously [8].

### 2.4. Preparation of 1-Hexadecanol as an Odorant

1-Hexadecanol is implicated as a female attractant and was prepared based on Zhang et al. [12]. For these experiments, 1-hexadecanol was purchased from Sigma-Aldrich (Burlington, MA, USA). 1-Hexadecanol was prepared as a 1M stock solution in ethanol and diluted to 100 μM in mineral oil. An amount of 1 mL of the mineral oil mixture was placed in a plastic dish between the wire cage top and the filter top. This permitted mice to smell the odorant while preventing physical access. For the investigation time experiment we prepared the 1-hexadecanol as above but diluted the stock solution to 100 μM in ethanol. Vehicle consisted of a 1:10,000 dilution of ethanol in mineral oil.

### 2.5. Collection and Preparation of Urine

For the urine testing condition, urine was obtained from a male mouse by scruffing the animal and collecting, with a kimwipe, freely secreted urine. The kimwipe from one male was placed on a separator positioned on the wire caging, in a plastic weigh boat, but below the cage filter top, so preventing physical contact with the mice while sealing in the scent. Females were placed into the cage and allowed to explore. After 1 h, a second vaginal moisture measurement was obtained and females were moved back into their home cage. A cage prepared in this manner was used only once (for one cage of females).

### 2.6. General Method of Exposure to Odorant (1-Hexadecanol)

For odorant exposure experiments, several cages containing a group of CD1 strain female mice (3–4/cage) were brought into an adjoining room separated by a closed door. Mice were tested in late-morning, starting at 10AM to reduce the likelihood of unusual scents. All baseline vaginal moisture readings were obtained for each mouse as described under Measurement of Vaginal Moisture above. This takes less than a minute per mouse so is accomplished in under half an hour. After this, the first cage of female mice were transferred into a scent-containing cage and allowed to explore. Subsequent groups of mice were transferred at ~4–5 min intervals. After an hour of exploration, the first group of mice were tested, with vaginal moisture readings taken for each female a second time to allow comparison to her own pre-exposure baseline. Subsequent cages were tested as they reached 1 h. Experimenters (both the experimenter obtaining a sample and the individual quantifying the distance that the sample traveled along the thread) were aware of the experiment that was being conducted (i.e., they were not blinded).

In a follow-on experiment, mice were allowed to physically interact with 1-hexadecanol. The experiment was conducted as above except that 1 mL of 100 μM 1-hexadecanol was added to a kimwipe that was mixed in with fresh bedding. During the hour of exploration females reliably came into physical contact with the 1-hexadecanol infused kimwipes, allowing for contact-exposure.

### 2.7. Measuring Investigation Time

Investigation time has a long history of use to assess olfactory discrimination [18]. For studies of investigation time, we strove to mimic the conditions of Zhang et al. since they reported positive findings with [13]. Fresh/clean empty cages were used (dimensions as above). On the distal ends of cages, in the center, circles (20 mm diameter) were drawn on the outside of the cage using faint blue marker to decrease visual salience of the mark. The circle was placed 5 cm from the cage bottom to allow ready accessibility to mice yet requiring some effort. For scent marking, 10 μL of solution was pipetted into the circle area on the inside of the cage and smeared evenly over the area using the pipette tip. Marks were given 5 min time to dry. At this time, the female mouse was placed into the cage, in the middle, and the top of the cage was covered with a clear, clean glass pane to prevent animal escape and other smells from intruding while offering full visibility to scorers. The mouse was allowed 5 min for exploration. Nose to circle area interactions were timed by two scorers with stopwatches, with one scorer for each end of the cage. Scorers alternated sides. As in Zhang et al., animals that did not interact with either of the circles were excluded from the statistical analysis. For the 1-hexadecanol experiment one of eight animals was excluded.

### 2.8. Method for Determining Estrous Phase

To determine the estrous cycle phase of a given mouse we used the method described by Byers et al. [19]. We obtained a vaginal smear by pipetting 10 μL of sterile saline into the proximal vaginal canal using a 20 μL pipette. The liquid was then collected back into the pipette and placed onto a slide and covered with a cover slip. The slide was then imaged within 3 h using a camera-equipped phase-contrast microscope. The picture was saved for scoring. Evaluation of the images was done via a two-person consensus approach: each person scored the images; in the case of disagreement, each made a case for a given phase and the scorers arrived at a consensus. Occasionally, scorers were uncertain about the phase because it exhibited characteristics of both. This occurred most frequently for estrus vs. metestrus. In that case, the sample was scored accordingly (e.g., estrus/metestrus) and the vaginal response data were excluded from analyses. These ambiguous estrous typings represented ~10% of the total.

Non-receptive estrous phases (i.e., metestrus, diestrus) were used as exclusion criteria for the repeated 1-hexadecanol vaginal response experiment (Section 3.3) to rule out the possibility that the non-response was due to a preponderance of non-receptive females in that experiment.

### 2.9. Statistics

Analyses were done using Graphpad Prism version 10. Experiments were analyzed using a two-tailed paired *t*-test comparing an experimental condition to the same-animal baseline. Detailed assessment of vaginal measurement data, including determination of sample size, was previously reported in [8].

## 3. Results

### 3.1. Vaginal Secretory Responses to the Scent of Dominant vs. Subordinate Males

In the initial experiments, female CD1 strain mice of indeterminate estrous phase were exposed to the urine of dominant or subordinate males, as described in Section 2. The urine of dominant males induced a vaginal secretory response (Figure 1A, paired *t*-test vs. baseline for dominant males, *p* = 0.046, t = 2.31, df = 9, *n* = 10), while the urine of subordinate males did not (Appendix A; *p* = 0.51, t = 0.69, df = 8, *n* = 9). In this initial experiment, the estrous state of the female mice was not monitored because we had previously found that the vaginal secretory response was sufficiently robust to be observed reliably in a general population of adult female mice. However, the negative finding for subordinate males could, in principle, have arisen if a large proportion of females were in their non-receptive state (e.g., [8]). We therefore repeated the experiment with a different cohort of females, now monitoring the estrous phase and including only data from females in their receptive phase (pro-estrus and estrus). We found that here too there was no response to subordinate males (Figure 1B; *p* = 0.31, t = 1.102, df = 7, *n* = 8).

### 3.2. Investigation Time of the Scents of Dominant vs. Subordinate Males

We confirmed previous findings [4] showing that females spend more time investigating the scent of dominant (DOM) vs. subordinate (SUB) males (Figure 2, paired *t*-test investigation time DOM vs. SUB: *p* = 0.025, t = 2.38, df = 20, *n* = 21).

### 3.3. Vaginal Secretory Responses to the Putative Female Attractant 1-Hexadecanol

As noted, we have previously reported that farnesenes elicit a robust vaginal secretory response in female mice of indeterminate estrus phase. Here we tested the response to a second candidate volatile pheromone 1-hexadecanol [12]. We exposed mice as described by Zhang et al. to closely approximate their method (described in Section 2) but found that hexadecanol did not elicit a vaginal secretory response (Appendix A, paired *t*-test vs. baseline: *p* = 0.14, t = 1.59, df = 13, *n* = 14). Because this initial finding was negative, we repeated the experiment with a different cohort of mice of known phase of estrus. We found that mice in their receptive phase (proestrus and estrus) still did not respond to the hexadecanol (Figure 3, paired *t*-test vs. baseline: *p* = 0.87, t = 0.1721, df = 12, *n* = 13). We additionally tested whether or not physical interaction with the hexadecanol would result in a positive response. Mice that were in their receptive phase did not experience a vaginal secretory response to hexadecanol (Appendix A, paired *t*-test vs. baseline: *p* = 0.063, t = 0.497, df = 8, *n* = 9).

### 3.4. Female Mice Do Spend More Time Investigating Hexadecanol

Our negative findings for vaginal secretory responses to hexadecanol could be explained if the mice were unable to smell the hexadecanol. We therefore tested investigation times as reported by Zhang et al., closely mimicking their experimental approach. We confirmed that females spend more time investigating hexadecanol than vehicle (Figure 4; paired *t*-test for time spent investigating hexadecanol vs. vehicle, *p* = 0.008, t = 3.90, df = 6, *n* = 7), indicating that they are able to detect the hexadecanol.

## 4. Discussion

Mice rely on chemical messengers to communicate a wide variety of information, including social status. We recently determined that female mice respond to male volatile pheromones with a vaginal secretory response, presumably as a preparatory response to the prospect of coitus. This offers a window into the physiological responses to pheromonal signals that we have here employed to examine whether or not this response is impacted by the male’s social status. We find that female laboratory mice experience a vaginal secretory response to dominant, but not to subordinate, males. In addition, while we had previously reported that the scent of chemical messengers α/β farnesenes induced a vaginal secretory response, here report that a second proposed female attractant, hexadecanol, does not.

When housed together, male mice and rats rapidly establish a social dominance hierarchy with profound physiological and behavioral consequences that also impact female responsiveness and, likely, mating success [4]. Chemical messengers may help females ascertain the social status of a given male, with evidence pointing to male preputial glands as a likely source of such chemical messengers. These glands undergo rapid morphological changes in response to male social status [6] and secrete both volatile and non-volatile compounds that communicate social hierarchy to conspecifics [10,20]. Females prefer to linger in spaces with higher preputial gland excretion content [21]. Several volatile and nonvolatile messengers have been actively investigated with several lines of evidence pointing to α/β farnesenes. The levels of farnesenes are higher in the preputial glands of dominant males, with levels shifting in response to changes in social status in as little as a week’s time [7]. α/β farnesenes therefore likely provide key information that females use to determine the social status of males. As noted, it is now possible to discern a physiological response–vaginal secretion–to male pheromones, and we have shown that farnesenes can elicit the vaginal secretory response [8].

But are there other chemical messengers that play the same role? And are there attractants that do not trigger a vaginal response? The study of attractants has been a subject of interest, not least for the pest-control industry, and a variety of additional candidate female attractants have been proposed, including the non-volatile darcin [14] and hexadecanol [11,13]. We chose hexadecanol here because, like the farnesenes, hexadecanol was described as volatile and produced by preputial glands [12,13]. In contrast to farnesenes, members of the terpene family that are known for their volatility, hexadecanol is a long chain fatty alcohol with a comparatively low vapor pressure [22], but our experiments suggest that mice are able to detect its scent. It is possible that only a subset of attractants also stimulate a vaginal response. This raises the question of what constitutes an attractant. The standard method of determining attraction has been to quantify the time spent investigating scents to assay the valence of those scents to rodents (e.g., [11]). In the present context, female mice that spent more time investigating a scent have generally been taken as revealing an attraction. But investigation time is also used to examine novelty, as in the novel object recognition assay (e.g., [23]). Duration of investigation is a behavior that may be subject to many influences. Attraction, novelty, threat, or simply the balance of chemical messengers may each merit investigation. And what is taken as attraction may in fact be interest. In this respect, a stimulated vaginal secretory response offers a more defined window into what a putative attractant is communicating. While 1-hexadecanol does not appear to communicate the prospect of coitus, it may well communicate other information and elicit different responses.

These findings additionally have a bearing on the preparation hypothesis that posits that females will automatically experience a vaginal secretory response when confronted with the likelihood of coitus (e.g., [9]). This is presumably at least partly to protect the reproductive apparatus but may also serve to provide a more welcoming environment (e.g., pH) for spermatozoa. We recently showed that female laboratory mice do not invariably respond to a stimulus that would ordinarily elicit a vaginal secretory response: they do not respond during their rest phase or during metestrus. This implies that a more nuanced preparation hypothesis may apply here, one that may rely in part on how the likelihood of coitus is assessed by the female. In many species there is a courtship process, and there is no reason to suppose that mice differ in this regard. Vocalizations outside the human auditory range likely play a role (reviewed in [24]), as do chemical messengers, as discussed above. It is possible that females do not experience a vaginal secretory response to the scent of a subordinate male because interaction with a subordinate male is unlikely to lead to mating. A study of the impact of dominance on mating success showed that subordinate males are less likely to initiate mating and do not even approach females in the first 15 min [4]. Vaginal preparation for mating in that instance would be a pointless expenditure of physiological resources since coitus is unlikely.

## 5. Conclusions

The ability to monitor vaginal secretory responses to encounters with males offers a window into how mice determine the likelihood of coitus and the chemical messengers that mediate some of these responses. Here we have shown that female mice experience a vaginal response to the scent of dominant but not subordinate males. In a second finding we find that, in contrast to farnesenes, a second candidate volatile preputial gland-derived chemical messenger, 1-hexadecanol, does not elicit a response. These findings narrow both the range of conditions and the chemical messengers that may elicit a coital preparatory response in laboratory mice.

## Figures and Tables

**Figure 1 biology-14-01041-f001:**
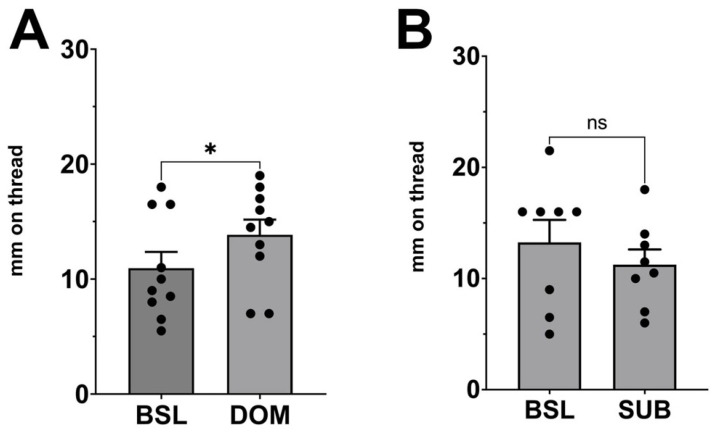
Vaginal secretory responses to dominant or subordinate males. Vaginal moisture was measured before (baseline, BSL) and after female CD1 strain laboratory mice were exposed to male urine. (**A**) Females responded to urine of dominant (DOM) males. (**B**) Females were non-responsive to the scent of subordinate (SUB) males even in a receptive phase of estrus. Bar graphs show mean + SEM. ns, not significant, *, *p* < 0.05 by paired *t*-test vs. same-animal baseline. *n* = 10, 8.

**Figure 2 biology-14-01041-f002:**
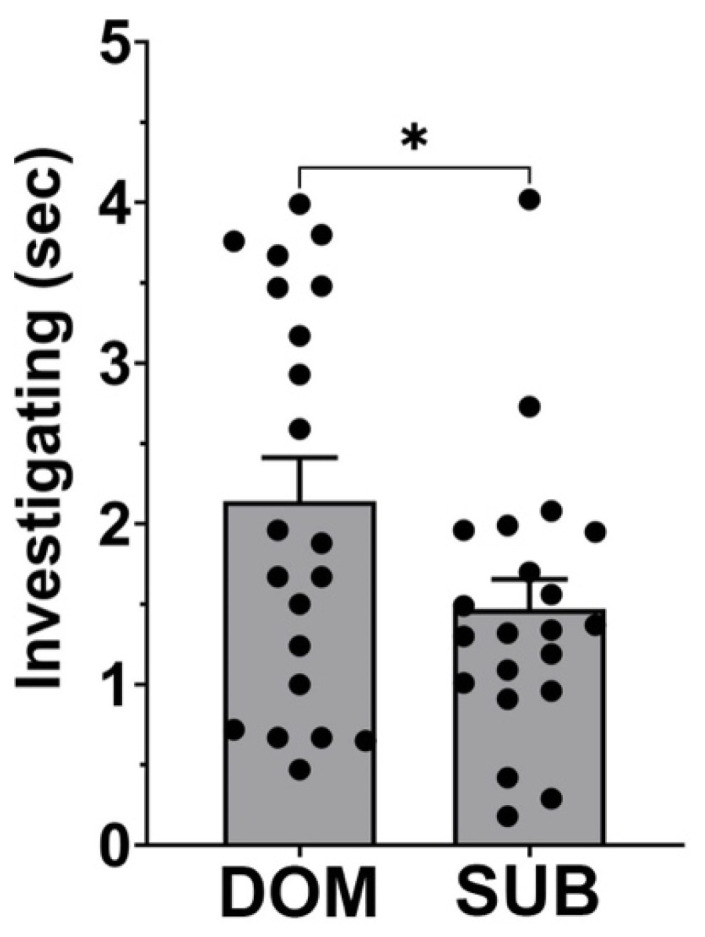
Investigation time of scents of dominant vs. subordinate males. Duration of investigation time (seconds) shows that female mice spend more time investigating urine of dominant (DOM) over subordinate (SUB) males. Bar graphs show mean + SEM. *, *p* < 0.05, paired *t*-test investigation time DOM vs. SUB: *p* = 0.025, *n* = 21.

**Figure 3 biology-14-01041-f003:**
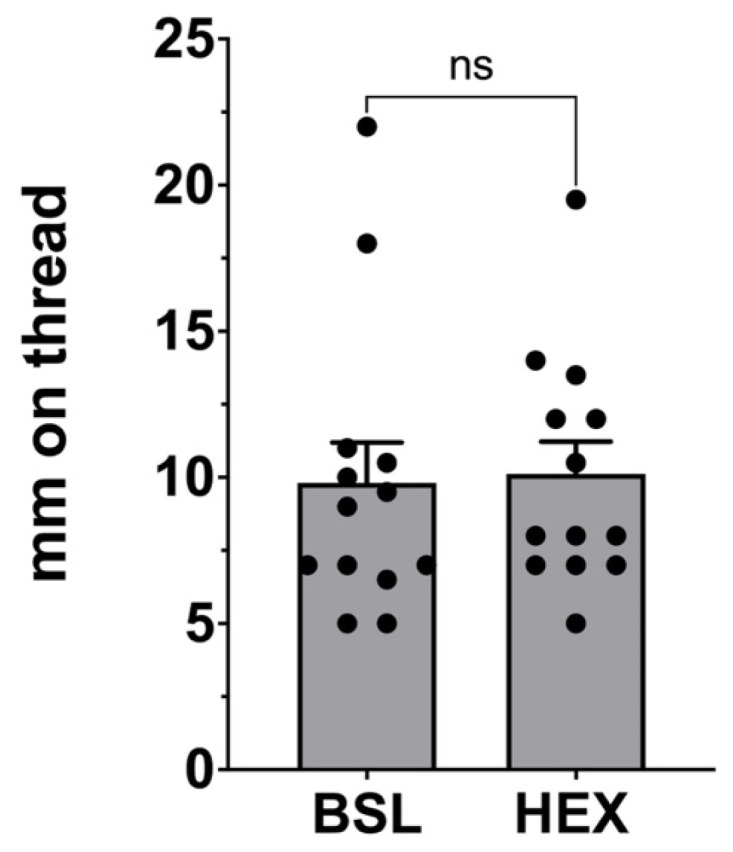
Vaginal secretory responses to the putative sexual attractant hexadecanol. In female mice in their receptive estrous phase, exposure to the scent of hexadecanol (HEX) for 1 h did not elicit a vaginal secretory response relative to baseline (BSL). Bar graphs show mean + SEM. ns, not significant. *p* = 0.87 by paired *t*-test vs. baseline, *n* = 13.

**Figure 4 biology-14-01041-f004:**
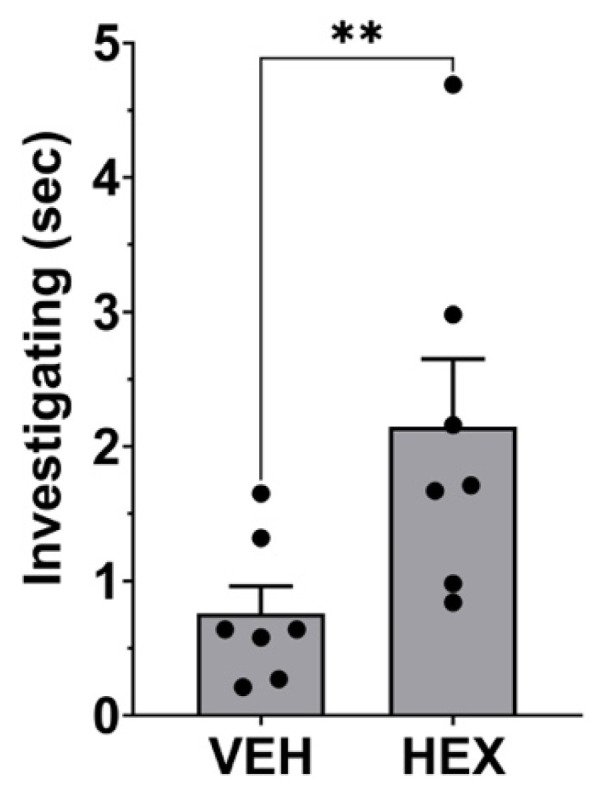
Investigation time of hexadecanol. Duration of investigation time (in seconds) is shown for vehicle and hexadecanol (100 μM). Bar graphs show mean + SEM. **, *p* < 0.01, paired *t*-test hexadecanol vs. vehicle, *n* = 7.

## Data Availability

Data that make up the figures in this manuscript will be included in a separate Appendix A.

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
