# Peer review of "The Impact of Male Social Status on Vaginal Secretory Responses in Mice"

_biology, 2025, doi:10.3390/biology14081041_

Round 1

Reviewer 1 Report (Previous Reviewer 3)

Comments and Suggestions for Authors

The authors conducted studies to examine whether vaginal responses to male scent vary according to the male's social status, and whether hexadecanol can elicit female vaginal responses. The experimental design, results, and writing are generally clear.

However, I have some major concerns regarding the novelty and significance of the findings.

Previous research has established that farnesenes induce vaginal responses and that dominant male mice possess higher levels of farnesenes compared to subordinate ones. Thus, it is not surprising that the odor from dominant males induces vaginal responses while that from subordinate males does not.

The primary new insight offered by this study is the assessment of hexadecanol as a potential female attractant. Unfortunately, the results were negative. While negative results are still valuable, they demand a more thorough explanation. The authors should provide more detail on why hexadecanol does not elicit a response despite its similarities to farnesenes.

Overall, the significance of the findings appears limited. However, the study's value could be enhanced if the authors tested multiple other putative female attractants, as mentioned in the introduction and discussion.

I recommend the authors consider expanding their research to include a wider range of potential attractants, which may warrant reconsideration of the study's impact.

Author Response

The authors conducted studies to examine whether vaginal responses to male scent vary according to the male's social status, and whether hexadecanol can elicit female vaginal responses. The experimental design, results, and writing are generally clear.However, I have some major concerns regarding the novelty and significance of the findings.

Previous research has established that farnesenes induce vaginal responses and that dominant male mice possess higher levels of farnesenes compared to subordinate ones. Thus, it is not surprising that the odor from dominant males induces vaginal responses while that from subordinate males does not.

Response #1: We wonder whether there was some miscommunication.  The reviewer made this point previously and we replied accordingly.  We include the response here again. 

We are somewhat bewildered by the reviewer’s argument.  There are many important experiments that demonstrated expected findings.   No one (hopefully) was surprised that a feather fell with the same acceleration as a hammer when dropped on the airless surface of the moon, but it was still an important experiment that had not previously been feasible. We agree that the result is expected based on the available evidence but not that the findings are therefore uninteresting or of limited significance.  A vaginal response has not previously been demonstrated because no assay was available for it.  Vaginal secretion offers an unambiguous physiological indicator. 

The experiment needed to be done (and to be published).  This is also a further demonstration of the utility of this model.

The primary new insight offered by this study is the assessment of hexadecanol as a potential female attractant. Unfortunately, the results were negative. While negative results are still valuable, they demand a more thorough explanation. The authors should provide more detail on why hexadecanol does not elicit a response despite its similarities to farnesenes.

Response #2: We do not assign a negative valence to the hexadecanol results.  Instead this was an opportunity to validate or call into question the assays (loitering time near a given candidate pheromone) previously used to determine what is an attractant.  We already discuss the issue at some length (Discussion, paragraph 3: Lines 456-467). 

Overall, the significance of the findings appears limited. However, the study's value could be enhanced if the authors tested multiple other putative female attractants, as mentioned in the introduction and discussion.  I recommend the authors consider expanding their research to include a wider range of potential attractants, which may warrant reconsideration of the study's impact.

Response #3: Again, unless we have missed an updated version of this reviewer’s comments, this is the point raised in response to the original submission.   As noted previously, we disagree regarding the significance of the findings and do not believe that the manuscript needs to be bolstered with additional experiments to add interest.   But there is a reason to consider the other attractants separately.  As noted at the very end of the introduction, the attractant we tested shared two important characteristics with the farnesenes, being both volatile and derived from preputial glands.  This means that these chemical messengers can act at a distance, a crucial distinction in behavioral ecology.  The other non-volatile attractants require physical contact to exert their effects and are therefore qualitatively different (putative) attractants that calls for separate investigation and interpretation. 

Reviewer 2 Report (Previous Reviewer 1)

Comments and Suggestions for Authors

The revision is acceptable. I have no further comments.

Author Response

The revision is acceptable. I have no further comments

Reply:  We thank the reviewer.

Reviewer 3 Report (New Reviewer)

Comments and Suggestions for Authors

Thank you for evaluation of the paper I reviewed the article entitled:The Impact of Male Social Status on Vaginal Secretory Responses in Mice

Line 26-34 in abstract section should be reorganized again with rephrasing .

The conclusion section  is not clear and not stated well please reorganized this part.

More information needed regarding the chemical messengers.

What about research hypothesis?

What about research gap and question,?

Please provide the aim of the work in simple form.

Material's 

How authors calculated sample size?

What about the light cycle system?

What about ethical approval number?

Please provide more about measurements of vaginal moisture???

Line 155 Measuring Investigation Time need more references and details 

Results are well designed 

Discussion is ok

Lines 285 -291 need more references and more details.

Conclusion is ok

Please make the refrences as journal style

Comments on the Quality of English Language

The paper need deep English language edition 

Author Response

Line 26-34 in abstract section should be reorganized again with rephrasing .

Response #1:  We have rephrased/reorganized that section and hope that the reviewer finds it to be improved.

The conclusion section  is not clear and not stated well please reorganized this part.

Response #2: We assume the reviewer is referring to the conclusions of the abstract section and have substantially revised the wording of this section.  Again we hope the reviewer considers it an improvement.

More information needed regarding the chemical messengers.

Response #3: Lines 98-99 state the key information for these compounds, that they are volatile and produced by preputial glands.  If the reviewer would like to see additional information about these compounds we will require some specific guidance as to what sort of details we should include.

What about research hypothesis? What about research gap and question,?

Response #4: We answer these together since they are related.   In the introduction, we have now added a sentence identifying the research gap and question.  We have added an explicit research hypothesis to the introduction.

Please provide the aim of the work in simple form.

Response #5: We added a single sentence to the opening of the last section of the introduction (lines 103-104) to summarize in a broad sense the overarching aim of the work.

Material's 

How authors calculated sample size?

Response #6: Details on how the sample size was calculated can be found in the original methods paper.  This is indicated under statistics in the methods section. 

What about the light cycle system?

Response #7: Details of the light cycle can be found on lines 106-11.

What about ethical approval number?

Response #8: The reviewer can find this detail on line 118.

Please provide more about measurements of vaginal moisture???

Response #9: Here too we point to the methods paper from last year that went into this in exhaustive detail including the method of manufacture of these threads.

Line 155 Measuring Investigation Time need more references and details 

Response #10: We have added a sentence indicating that we were endeavoring to mimic the experimental design of Liu et al.  and have added some additional detail.

Results are well designed; Discussion is ok. Conclusion is ok

Response #11: We thank the reviewer.

Lines 285 -291 need more references and more details.

Response #12: unfortunately in our version these lines are in the results section.  We will require more information about which section the reviewer is referring to.

Please make the refrences as journal style

Response #13: We have changed the references to MDPI style. 

Round 2

Reviewer 1 Report (Previous Reviewer 3)

Comments and Suggestions for Authors

Thank the authors for their revision. However, I noticed that my initial concerns remain unaddressed. The novelty of the findings still limited, particularly regarding the lack of responses to hexadecanol and the significance of the comparisons between dominant and subordinate males. It would greatly strengthen the paper if the authors provided a more detailed analysis of why hexadecanol did not elicit a response, and I still urge to explore additional potential female attractants to enhance the study's overall impact.

Author Response

Thank the authors for their revision. However, I noticed that my initial concerns remain unaddressed. The novelty of the findings still limited, particularly regarding the lack of responses to hexadecanol and the significance of the comparisons between dominant and subordinate males.

A: We still contest this assessment.   None of the other three reviewers have expressed any concerns about the novelty or significance of these findings.   Indeed, we refer to reviewer #2’s comment in response to our initial submission in May:  “The authors’ earlier paper (Sci. Rep. 2024, 14) is, according to my opinion, groundbreaking in the way that it describes a method for obtaining a measure of the vaginal response in rodents. In the present paper, they extend these observations. I would strongly recommend the authors to emphasize that they present unique data of utmost importance concerning the quantification of rodent sexual arousal in the form of a vaginal response. It appears that the authors themselves are unaware of the importance of their methods.” 

It would greatly strengthen the paper if the authors provided a more detailed analysis of why hexadecanol did not elicit a response.

A: We have added some discussion of differences between hexadecanol vs. farnesenes and have additionally added an experiment testing whether physical interaction with hexadecanol might elicit a vaginal secretory response.

I still urge to explore additional potential female attractants to enhance the study's overall impact.

A:  As noted in our previous reply, the other candidate attractants are non-volatile and are therefore qualitatively different since they do not allow for perception at a distance.  We intend to investigate these as part of a separate follow-on study, but this will require separate methods and the assistance of a synthetic chemist.  

This manuscript is a resubmission of an earlier submission. The following is a list of the peer review reports and author responses from that submission.

Round 1

Reviewer 1 Report

Comments and Suggestions for Authors

The manuscript reports a very interesting study of the vaginal response to olfactory attractants in mice. Odors from dominant and subordinate males were compared for their capacity to stimulate vaginal secretion. It was found that only urine from dominant males caused a response. A supposedly attractive odorant, hexadecanol, had no effect.

The manuscript is well written, the methods are excellent, the statistical analysis is correctly performed, the results are clear, and the discussion is reasonable. I have only a few minor comments.

In women, the vaginal response to sexually relevant stimuli is considered an exquisite indicator of sexual arousal. Such responses have been used in hundreds of studies, and they have provided a large amount of basic information concerning female sexual responses. The authors’ earlier paper (Sci. Rep. 2024, 14) is, according to my opinion, groundbreaking in the way that it describes a method for obtaining a measure of the vaginal response in rodents. In the present paper, they extend these observations. I would strongly recommend the authors to emphasize that they present unique data of utmost importance concerning the quantification of rodent sexual arousal in the form of a vaginal response. It appears that the authors themselves are unaware of the importance of their methods. In relation to this, the discussion would benefit from being focused on sexual arousal rather than on speculations about the role of dominance for reproductive success, or about the outdated “preparation hypothesis”.

In women, the vaginal lubrication is a transudate. How do the authors know that it is a result of secretion in mice? Secretion from where?

The title, “..murine vaginal…..” is overly ambitious. Why not simply “….vaginal secretions in mice….”.

I would also recommend the authors to use hormone-primed, ovariectomized females in future studies. This would allow for a much better control of the endocrine status than rude classification of stage of the estrous cycle used in the present study.

What is illustrated in the figures? Means, medians, standard errors, standard deviations, interquartile ranges?

Investigation time is not a sexual response per se, as the authors point out. To the contrary, the vaginal response is specific to sexually relevant stimuli, at least in women. Perhaps the data on investigation time are entirely irrelevant. At least they do not offer any useful information. 

Reviewer 2 Report

Comments and Suggestions for Authors

General Evaluation

This manuscript addresses an interesting and potentially significant question concerning the physiological effects of male social status on female reproductive responses in mice. However, despite its conceptual appeal, the study suffers from fundamental methodological flaws and interpretive overreach that severely limit its current value. Without substantial revision and additional controls, the conclusions drawn from these experiments are not well supported by the data.

Major Points

1.  Inadequate Definition of Social Dominance

The classification of male mice as "dominant" or "subordinate" based solely on weight differences within group-housed cages is scientifically unjustifiable. This method lacks both empirical validation and behavioral grounding. The cited reference (14) explicitly argues against such use of body weight as a proxy for social status. Without direct behavioral assessments of dominance (e.g., aggressive interactions, scent-marking, or other ethologically valid markers), the foundational grouping of the study is unreliable, which seriously undermines the integrity of all related findings.

2.  Description of Vaginal Moisture Measurement

The core measurement of vaginal secretory response is novel, yet inadequately documented. Including images of the capillary and the phenol red thread after exposure would significantly improve reproducibility. Additionally, further explanation is needed regarding the 10-second insertion duration—e.g., whether it is sufficient to reach saturation, and how this duration may affect measurement variability.

3.  Ambiguity in Use of Stimulus Males

The manuscript fails to clarify how many individual males were used, how frequently each was tested, or whether the same individuals repeatedly served as "dominant" or "subordinate." This omission is not trivial—it raises questions about potential pseudoreplication and limits the generalizability of the results.

4.  Estrous Phase Assessment and Integration
While the estrous cycle phase is assessed via smear, the methods section does not clearly explain how this information was used in data selection or analysis. Rather than reporting the experimental process in chronological order, the manuscript should present the finalized experimental design with clear criteria for inclusion/exclusion of data based on estrous phase.

5.  Vehicle Control in Hexadecanol Preference Tests

In the odor preference tests, the description of the vehicle used with hexadecanol is insufficient. Furthermore, it is questionable whether increased investigation time relative to a neutral vehicle can be interpreted as attraction, especially when HEX is a volatile cue and the vehicle is likely odorless or no salient odor.

6.  Disconnection Between Male-Based and Compound-Based Tests
There appears to be a conceptual and methodological gap between the initial experiments using urine from dominant/subordinate males and the later experiments testing isolated chemical stimuli (hexadecanol). Without confirming the actual levels of hexadecanol or α/β farnesenes in the urine samples, the authors cannot confidently claim these molecules mediate the observed differences. Additional validation—such as supplementing subordinate urine with these molecules—would strengthen the causal interpretation.

Minor Points

  • The abstract should briefly describe the methods used to measure vaginal responses and prepare stimulus males.
  • Figure 1A (uninformed by estrous phase) may be better placed in the Supplementary Material, with the main focus on data from estrus-confirmed females.
  • A comparison between estrus and non-estrus conditions would provide valuable insight into the interplay between internal state and responsiveness.

Conclusion

This manuscript presents interesting and potentially impactful findings, but critical revisions are needed—particularly regarding experimental design, clarity of methods, and the interpretation of behavioral versus physiological responses. Addressing the above points would significantly enhance the scientific rigor and interpretability of the work.

Reviewer 3 Report

Comments and Suggestions for Authors

The authors conducted studies to examine whether vaginal responses to male scent vary according to the male's social status, and whether hexadecanol can elicit female vaginal responses. The experimental design, results, and writing are generally clear. However, I have some major concerns regarding the novelty and significance of the findings.

Previous research has established that farnesenes induce vaginal responses and that dominant male mice possess higher levels of farnesenes compared to subordinate ones. Thus, it is not surprising that the odor from dominant males induces vaginal responses while that from subordinate males does not.

The primary new insight offered by this study is the assessment of hexadecanol as a potential female attractant. Unfortunately, the results were negative. While negative results are still valuable, they demand a more thorough explanation. The authors should provide more detail on why hexadecanol does not elicit a response despite its similarities to farnesenes.

Overall, the significance of the findings appears limited. However, the study's value could be enhanced if the authors tested multiple other putative female attractants, as mentioned in the introduction and discussion.

I recommend the authors consider expanding their research to include a wider range of potential attractants, which may warrant reconsideration of the study's impact.
